

Materials Science

# Effect of oxide scale structure on shot-blasting of hot-rolled strip steel

Xiaochen Wang[1], Rui Ai[1], Quan Yang[1], Shang Wang[2], Yanjie Zhang[1], Yingying Meng[1] and Xianghong Ma[3]

[1] School of Engineering Technology Research, University of Science and Technology Beijing, Beijing, China
[2] School of Automotive Engineering, Beijing Polytechnic, Beijing, China
[3] School of Engineering and Applied Science, Aston University, Birmingham, UK

## ABSTRACT

**Background:** The effect of oxide scale composition of hot-rolled strip (Q235) on shot blasting is studied in this article. The properties of the oxide scale on the strip surface change during storage. The shot blasting is an important on-line acid-less descaling technology. The effect of shot blasting is affected by many factors, among which the composition of oxide scale may play an important role. However, there are few studies on the relationship between the oxide layer content and the descaling effect.
**Methods:** The morphologies of oxide scales at different storage times are observed by scanning electron microscopy (SEM), and the compositions are analyzed by X-ray diffraction. These strips are then shot blasted and descaled with different amounts of abrasive, and the descaling effects are compared by SEM.
**Results:** The results show that the eutectoid structure $Fe_3O_4/Fe$ in the oxide scale will gradually transform into $Fe_3O_4$. In the case of short storage time, the content of the eutectoid structure is high, and it is difficult to remove the oxide scale. While the strip with a long storage time has no eutectoid structure $Fe_3O_4/Fe$ and FeO, it is easy to remove the oxide scale during the shot blasting process. The composition of the oxide scale has a significant effect on the effect of shot blasting, and it provides significant guidance to the optimization of the descaling process parameters.

## INTRODUCTION

During the steel strip rolling and cooling process, a dense and brittle oxide scale will form on the surface. Before the further cold rolling or galvanizing process, the oxide scale is usually removed by pickling to ensure the surface quality of the finished product (*Sun et al., 2003*; *Bin, Guang-ming & Zhen-yu, 2010*).

Due to the serious environmental pollution problem caused by the pickling process, scholars have long been committed to the research of acid-free descaling technology to replace the pickling process, and have achieved valuable theoretical and experimental achievements. The principle of high-pressure water descaling is to use high-pressure pumps to generate high-pressure water. The high-pressure water jets cause thermal changes, shocks, vibrations, and scouring on the surface of the strip. The dynamic pressure of high-pressure water becomes the hydrostatic pressure and invades the bottom of the

Corresponding author
Xiaochen Wang,
wangxiaochen@ustb.edu.cn

oxide scale, causing the oxide scale to peel off from the surface of the substrate (*Choi & Choi, 2002*). This technology is widely applied in hot rolling process, but it can't be used to the cold rolling procedure since the energy of water is too small to remove scales.

Abrasive water jet descaling technology uses high-pressure water to accelerate steel sand, quartz sand and other discrete bodies, and sprays the mixed abrasive stream to the strip surface at a certain angle through a nozzle to crush the oxide scale. Both the discrete body and water in this method can be recycled, and the descaling effect is obvious. However, due to the large water flow of the system, the high-pressure plunger pump requires higher cleanliness of the water, the water circulation system is always in a high load state, and the nozzle wears severely under long-term service, so this technology can only be applied to narrow band steel descaling or surface treatment of small parts (*Meng, Wei & Ma, 2016*).

Tensioning descaling is a mechanical method of repeatedly bending the strip steel. After the metal substrate is subjected to stress, a certain elastoplastic deformation occurs. The oxide scale on the surface of the metal substrate is broken due to brittleness and the purpose of descaling is achieved. Tensioning descaling is generally used in cases where the material is not seriously hardened and the product quality requirements are not strict (*Tongqing, 1998*; *Bakhmatov et al., 2014*). Smooth-Clean Surface technology is used in a closed space to automatically adjust the roll gap of the grinding roller according to the thickness of the strip. At the same time, the surface of the steel plate is continuously washed by circulating filtered water, and the ground iron oxide is taken away to achieve surface cleaning. Finally, a seven μm thick anti-rust layer is formed on the surface of the metal substrate. This method is not suitable for cold rolling, deep drawing and rotary deep drawing (*Tamura et al., 2020*).

In order to realize the application of on-line acidless descaling for broad steel strip production, the Material Works Ltd. of the USA developed the EPS (Eco-Pickled Surface) system. In this system, the abrasive shot blasting was applied to the industrial fields and proved to be an effective method to ensure the strip surface quality after descaling (*Voges & Mueth, 2012*; *Voges, Mueth & Lehane, 2008*). However, the energy consumption and processing cost of the system is so high that it cannot replace the pickling process yet.

Our research group is very interested in the non-acid oxide scale removal technology, and proposed the oxide scale removal technology combining shot blasting with high-pressure water, and designed the relevant experimental equipment. The most important research direction is to optimize the process parameters of shot blasting to reduce system energy consumption and processing costs. We have studied the effect of shot blasting speed on the descaling effect, and the results show with the increase of the projectile velocity, the damage area of the oxide scale is increased, and the damage area is composed of the direct destruction area and the indirect failure area. (*Wang et al., 2017a*, *2017b*, *2018*).

Actually, the effect of shot blasting is affected by many factors, among which the composition of oxide scale also has an important effect on the removal of scales. Parameters such as steel type, rolling speed, rolling temperature, cooling speed and coiling temperature, etc. will affect the oxide scale composition (*Zhou et al., 2011*; *Gong et al.,*

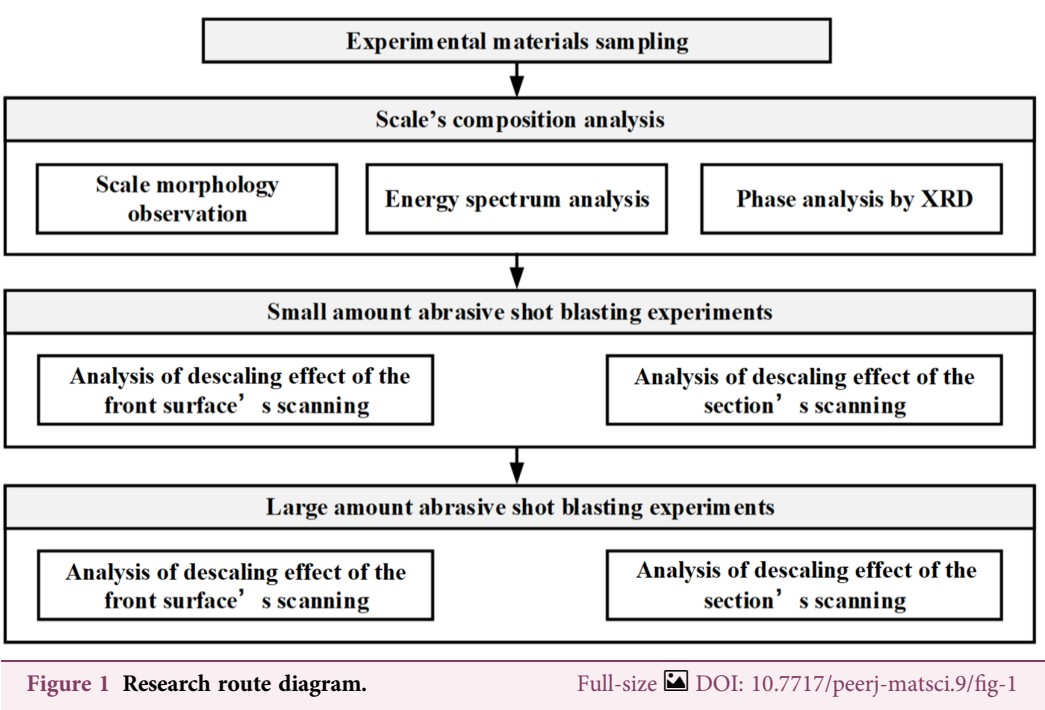

**Figure 1  Research route diagram.**

*2009*). The oxide scale layer of ordinary carbon steel generally consists of three layers (*Bonnet et al., 2003*): the inner layer is a solid solution of FeO and $Fe_3O_4$, the middle layer is $Fe_3O_4$, and the outer layer is $Fe_2O_3$. During the hot rolling process, the main component of the oxide scale layer is FeO. According to the Fe-O equilibrium phase diagram (*Chen & Yuen, 2000*, *2002*, *2003*), the eutectoid reaction of FeO can produce a mixed product of Fe and $Fe_3O_4$ below 570 °C. After laminar cooling and air cooling, a large amount of FeO will transfer into precipitates by eutectoid reaction. After being exposed to the air for a long time, the outermost layer of the oxide layer continues to be oxidized to $Fe_2O_3$. Therefore, in the process of the exposure in the air, the oxide scale composition is varying continually. However, there are few studies on the relationship between the oxide layer content and the descaling effect, which is a key factor affecting the descaling effect, and is also the research objective of this article.

   In this article, two kinds of Q235 strip steels with different air cooling time are selected for the research. Firstly, the difference in scale composition on the strip surface is obtained through energy dispersive spectrometer (EDS) and X-ray diffraction (XRD) analysis. Then the shot blasting experiments are carried out, and the descaling effect is observed by scanning electron microscopy (SEM). Moreover, the influence of the variation of the scale's composition caused by air cooling time on the descaling effect of shot blasting is analyzed, which provide important guidance to the improvement of acidless descaling process in industrial production. The research route is shown in Fig. 1.

## MATERIALS AND METHODS

The test samples are two Q235 strips stored at different times. Firstly, the oxide scale morphologies and compositions are measured by EDS and XRD. Then, the descaling

**Table 1 The chemical composition of the samples.**

| Sample no. | Fe/% | C/% | Mn/% | Si/% | S/% | P/% |
|---|---|---|---|---|---|---|
| No. 1 | >97 | 0.17 | 0.31 | 0.15 | 0.03 | 0.020 |
| No. 2 | >97 | 0.19 | 0.26 | 0.13 | 0.028 | 0.017 |

**Notes:**
This figure shows the chemical composition of the two samples.
**DO1:** Effect of oxide scale composition of hot-rolled strip steel on shot blasting/table-1.

experiments are performed using the shot blasting descaling experimental device developed by National Engineer Research Center of Flat Rolling Equipment (NERCFRE). The electron microscope is used to observe the removal effect in the experiments.

The object of XRD inspection is the surface of the sample after shot blasting. The bottom of the sample is fixed on the platform by means of bonding. The model of the XRD device is Ultima IV, and the type of tube is ceramic X-ray tube.

As for SEM, the sample was cut into 10 mm × 10 mm squares and then embedded into the resin. The SEI mode is used to observe the surface morphology, and the BSEI mode is used for element detection. The type of equipment used is ULTRA 55.

## Experimental materials

The experimental samples were taken from the Q235 hot rolled strip of the practical production line of a steel company. During the hot rolling process, the temperature dropped from 1,050 °C to 870 °C. Both the samples are 1 m × 1 m in size and 3 mm in thickness. One of the samples was produced 1 year ago and stored in the room environment, the sample and the group of subsequent specimens from which were labeled as No. 1. The other was produced within 1 week before the experiment, the sample and the group of subsequent specimens from which were labeled as No. 2. The reason for using samples that have been stored for a long time is to explore the descaling effect of samples with different oxide layer compositions, and the composition of the oxide layer can also be changed in other ways. Table 1 shows the chemical composition of the two samples. It can be seen there are little difference between them, and the influence on the mechanical properties can be neglected.

## Experimental method
### Oxide scale's composition analysis

The procedure was as follows:
(a) Both sample No. 1 and No. 2 were cut in the middle area to obtain eight specimens with size of 20 mm × 10 mm respectively. The surface of the sample was cleaned by ultrasonic cleaner, then wiped with alcohol and dried with a dryer.
(b) Four specimens from both No. 1 and No. 2 groups were made into mounts with the cutting surface as the front side and polished, respectively. Then the oxide scale morphology observation and EDS were conducted by the ZEISS ULTRA 55 scanning electron microscope.
(c) Other four specimens from both No. 1 and No. 2 groups were taken to the phase analysis by the Ultima IV X-ray diffractometer, respectively.

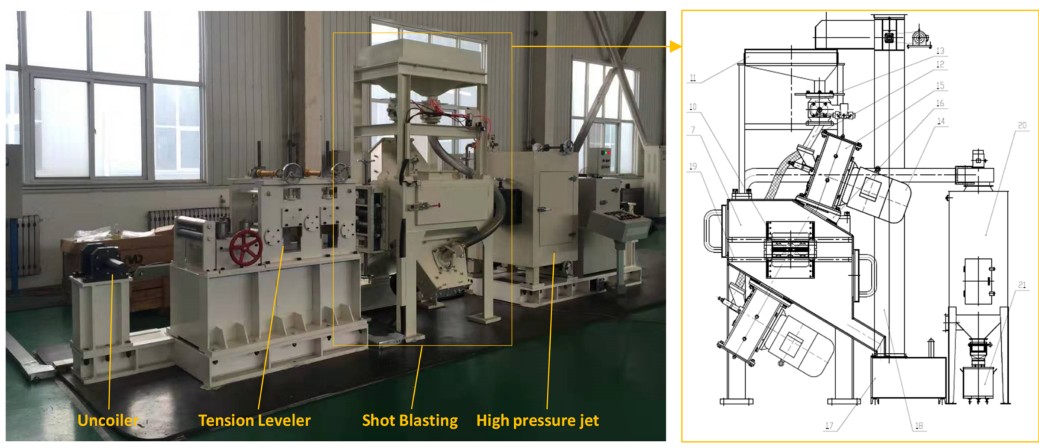

**Figure 2 Acidless descaling experimental facility.** The acidless descaling experimental facility designed by NERCFRE.

### Shot blasting experiments

**(1) The slot blasting descaling experimental facility**

The acidless descaling experimental facility designed by NERCFRE is shown in Fig. 2. This device mainly includes six major units, which are uncoiler, 5-roller tension leveler, slot descaling, high-pressure jet, sweeping-drying and coiler. The main process parameters include: impact angle, impact line speed, particle size and abrasive weight.

**(2) The shot blasting experiments with a small amount of abrasive**

(a) Both sample No. 1 and No. 2 were cut in the middle area to obtain one specimen with size of 200 mm × 200 mm respectively. The surface of the sample was cleaned by ultrasonic cleaner, then wiped with alcohol and dried with a dryer.

(b) The slot blasting experiments were performed by the acidless descaling experimental facility. The impact angle θ is set to 60°, the impact line speed $v$ is 40 m/s, the particle size $D$ is 0.6 mm, and the abrasive weight $W$ is 2 kg.

(c) Both the samples in step b were cut in the middle area to obtain two specimens with size of 20 mm × 10 mm respectively, and the specimens obtained by cutting were divided into two groups. One group of specimens was observed by a ZEISS ULTRA 55 scanning electron microscope for the descaling effect on the front of the specimens. Another group of specimens was mounted with the cut surface, and the descaling effect from the cut surface was observed.

**(3) The shot blasting experiments with a large amount of abrasive**

In order to analyze whether a sufficient amount of abrasive can remove the scales clearly, a total weight of 20 kg of abrasive was used in the experiment, and the remaining parameters were unchanged.

## RESULTS

### Results of scale composition experiments

The cross-section morphologies of the oxide scale observed by SEM are shown in Fig. 3. The EDS of the iron and oxygen elements at the outside, intermediate and inside

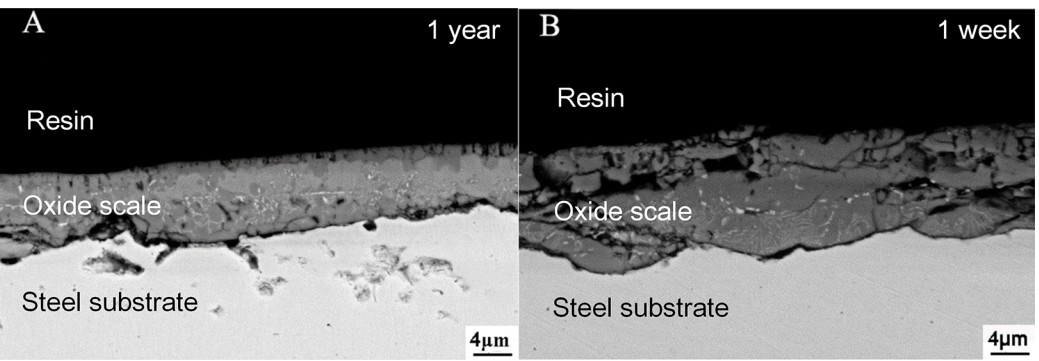

**Figure 3 The morphology of the oxide scale observed by SEM (A) No. 1 group (B) No. 2 group (Work Distance (WD) = 12.5 mm).** The cross-section morphologies of the oxide scale observed by SEM is shown.

**Table 2 Energy dispersive spectrometer of iron and oxygen elements of the oxide scale.**

| Positions of scale | $\omega_{Fe}$ | | $\omega_{O}$ | |
|---|---|---|---|---|
| | No. 1 | No. 2 | No. 1 | No. 2 |
| Outside | 76.34 | 85.10 | 23.67 | 14.90 |
| Intermediate | 76.96 | 85.77 | 23.04 | 14.23 |
| Inside | 79.50 | 85.21 | 20.50 | 14.19 |

Notes:
The energy dispersive spectrometer of the iron and oxygen elements at the outside, intermediate and inside positions of the oxide scale by the ZEISS ULTRA 55 scanning electron microscopy are shown in this figure.
**DO2:** Effect of oxide scale composition of hot-rolled strip steel on shot blasting/table-2.

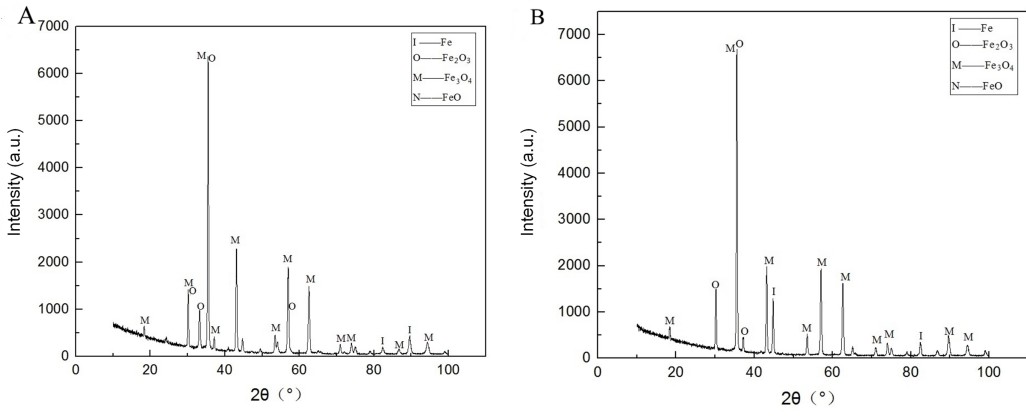

**Figure 4 The phase analysis by X-ray diffractometer (A) No. 1 group (B) No. 2 group.** The results of the phase analysis by X-ray diffractometer are shown.

positions of the oxide scale by the ZEISS ULTRA 55 scanning electron microscopy are shown in Table 2. The results of the phase analysis by X-ray diffractometer are shown in Fig. 4.

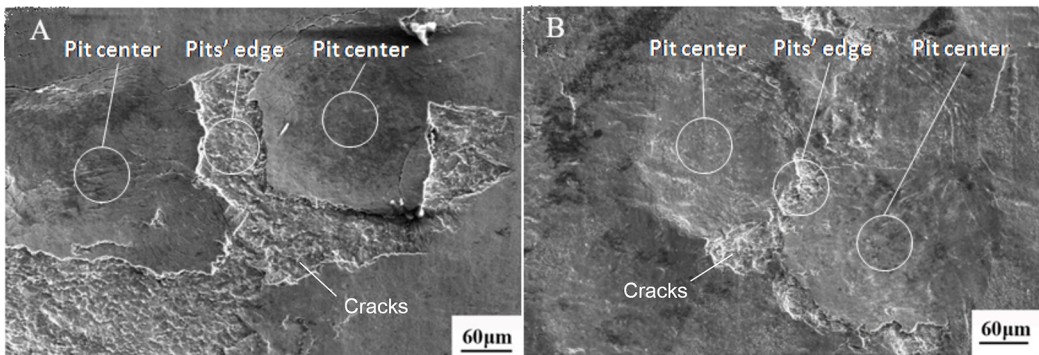

**Figure 5 The descaling effects from the front surface's scanning after the shot blasting with a small amount of abrasive (A) No. 1 group (B) No. 2 group (WD = 12.5 mm).** The descaling effects from the front surface's scanning after the shot blasting with two kg of abrasive are shown.

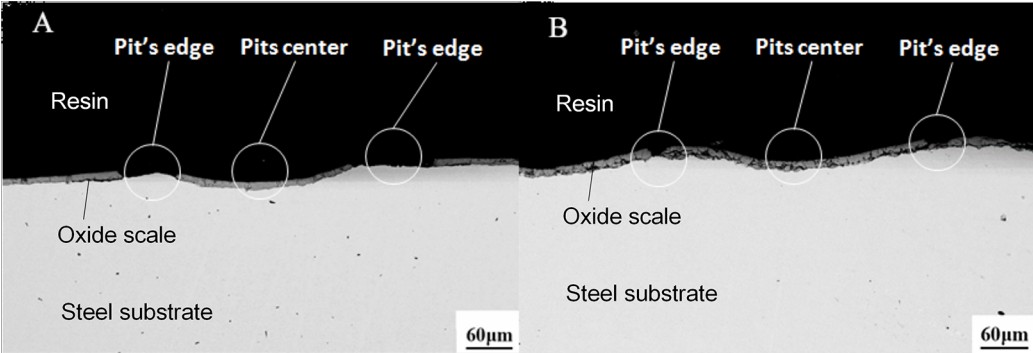

**Figure 6 The descaling effects from the front section's scanning after the shot blasting with a small amount of abrasive (A) No. 1 group (B) No. 2 group (WD = 12.5 mm).** The effect of descaling from the cross section is shown.

## Experiments of the shot blasting experiments with a small amount of abrasive

The descaling effects from the front surface's scanning after the shot blasting with two kg of abrasive are shown by Fig. 5.

The oxide scales are layered and have a certain thickness. It is difficult to determine whether the oxide scale is completely peeled off from the base body only from the front surface's scanning. Therefore, it is necessary to observe the effect of descaling from the cross section, as shown by Fig. 6.

The SEM results of No. 1 and No. 2 groups after shot blasting with 20 kg abrasive are shown in Figs. 7 and 8.

## Experiments of the shot blasting experiments with a large amount of abrasive

The 4,000 times magnifications of descaling effect from the section's scanning after the shot blasting with 20 kg of abrasive for No. 1 and No. 2 group are shown in Figs. 9 and 10.

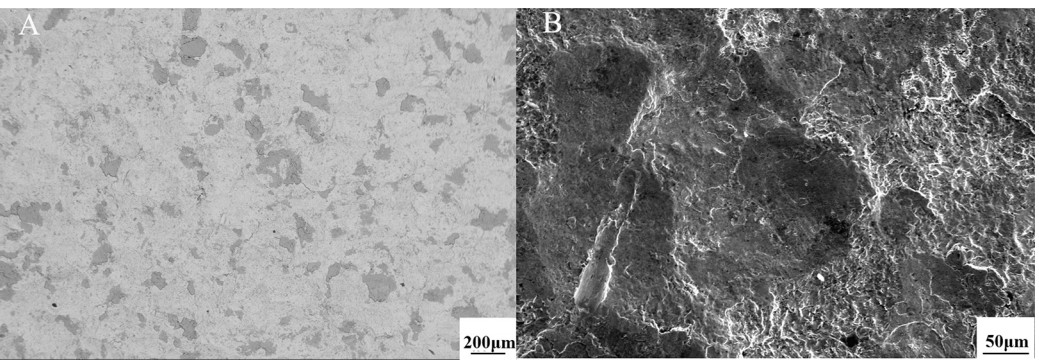

**Figure 7 The descaling effects from the front section's scanning after the shot blasting with a large amount of abrasive of No. 1 group (A) 100 times magnification (B) 500 times amplification (WD = 16 mm).** The SEM results of No. 1 group after shot blasting with 20 kg abrasive are shown.

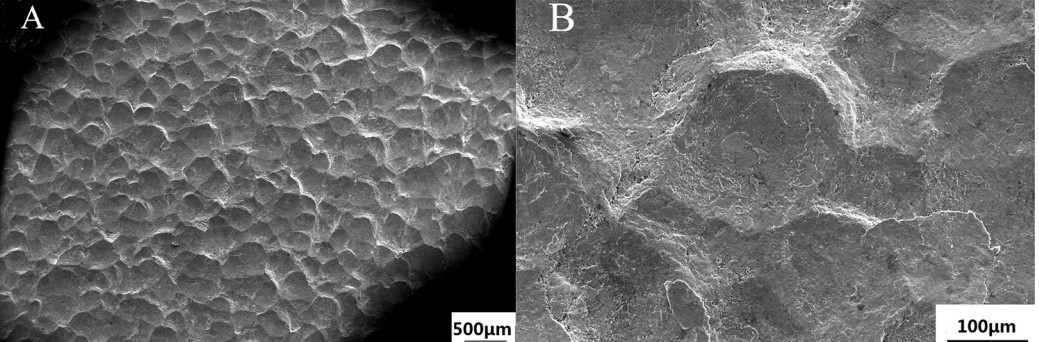

**Figure 8 The descaling effects from the front section's scanning after the shot blasting with a large amount of abrasive of No. 2 group (A) 50 times magnification (B) 500 times amplification (WD = 12.4 mm).** The SEM results of No. 2 group after shot blasting with 20 kg abrasive are shown.

Since Fig. 9 is the 4,000-times magnification result of the SEM, the field of view is very small. In order to improve the reliability of the research, a larger view field was chosen and the area scanning of EDS was conducted, as is shown in Fig. 11. Similarly, a larger view field was chosen and the area scanning of EDS for No. 2 group was conducted, as is shown in Fig. 12.

## DISCUSSION

### Scale composition analysis

#### Scale morphology analysis

As is shown in Fig. 3A, for the No. 1 group, the thickness of oxide scale is relatively uniform and is about 9.5 μm, and the structure is compact and well combined with the basal body, which indicates that the oxidation of the strip surface is uniform and adequate during the hot-rolling and long-time air cooling process. In Fig. 3B, for the No. 2 group, the uniformity of the oxide scale thickness is worse than that of No. 1 and the average

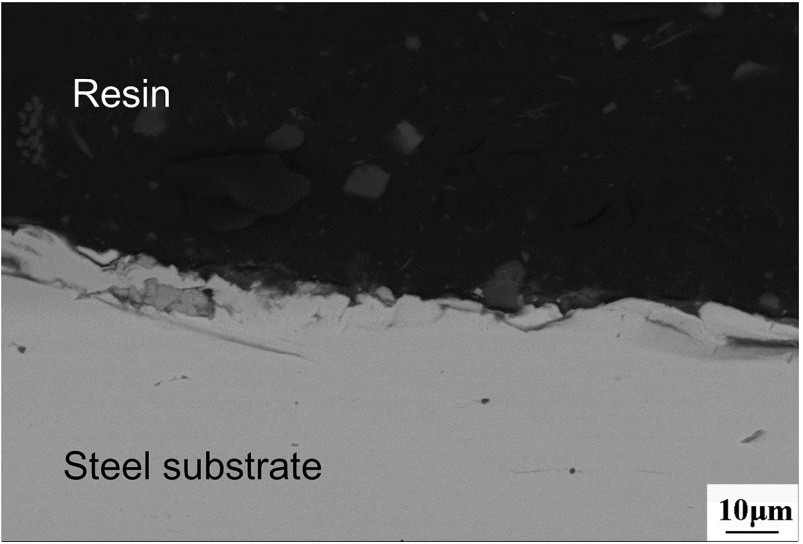

**Figure 9 The descaling effects from t he section's scanning after the shot blasting with a large amount of abrasive of No. 1 group (WD = 12.4 mm).** The 4,000 times magnification of descaling effect from the section's scanning after the shot blasting with 20 kg of abrasive for No. 1 group is shown.

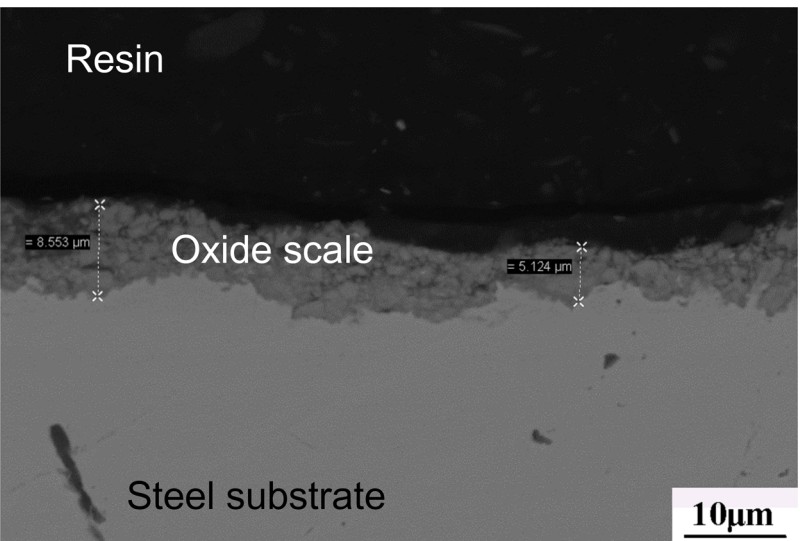

**Figure 10 The descaling effects from the section's scanning after the shot blasting with a large amount of abrasive of No. 2 group (WD = 12.4 mm).** The 4,000 times magnification of descaling effect from the section's scanning after the shot blasting with 20 kg of abrasive for No. 2 group is shown.

thickness is about 12 μm. It is obvious that there are many defects in the structure of oxide scale. By the above comparison, there are apparent differences of scale morphology with the rolling and cooling conditions difference. And as the chemical composition changes, the density of the oxide scale gradually increases.

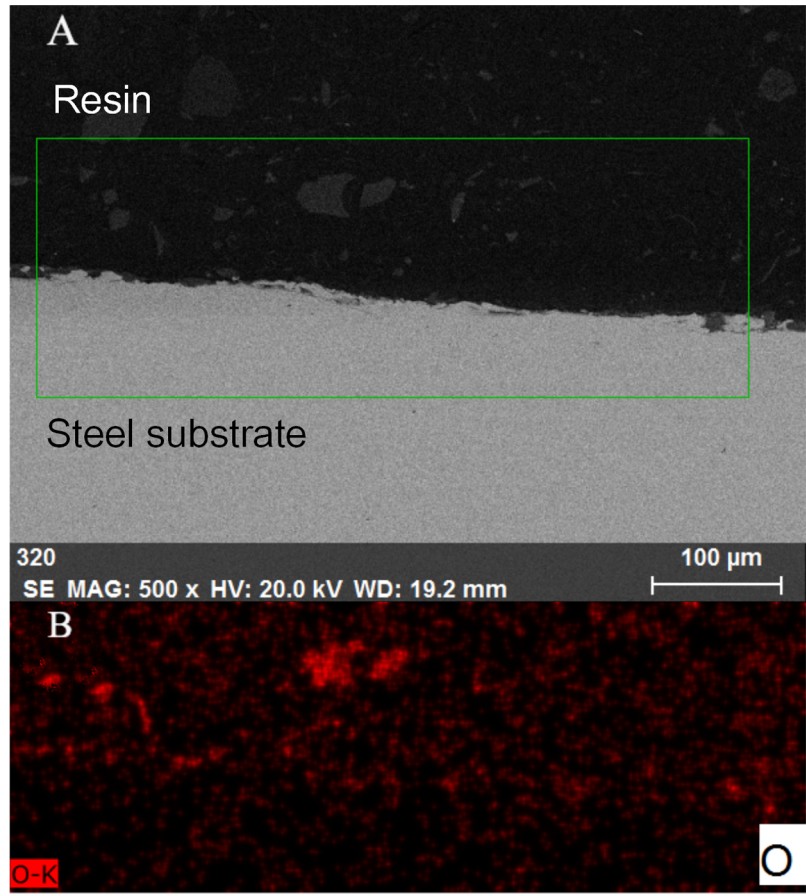

**Figure 11 Results of oxygen element scanning of No. 1 group (WD = 19.2 mm).** (A) The part in the green frame is the area scan object; (B) oxygen scan results. A larger view field chosen for the area scanning of energy dispersive spectrometer for No. 1 group is shown.

### The energy dispersive spectrometer

As is shown in Table 2, the values are the average of multiple measurements. The results show that there is almost no difference in the iron and oxygen content at different positions of the oxide scale for each group. In addition, the content of oxygen element at all positions of the oxide scale of No. 1 group is higher than that of the No. 2 group, which indicates the different oxidation effect caused by the storage time in the air.

### The phase analysis of scale

The diffraction peaks are identified according to the PDF2004 standard card. As is shown in Fig. 4A, the phase composition of the oxide scale is mainly $Fe_3O_4$ and $Fe_2O_3$ for the No. 1 group, and FeO and Fe are almost absent. It indicates that FeO is converted into $Fe_3O_4$ and Fe by the eutectoid reaction, and the eutectoid structure $Fe_3O_4$/Fe is oxidized to $Fe_3O_4$ subsequently during the long-time storage in the air. Thus, the scale's composition is mainly $Fe_3O_4$ with a small amount of $Fe_2O_3$. As is shown in Fig. 4B, for the No. 2 group, the phase composition of the oxide scale is mainly $Fe_3O_4$, $Fe_2O_3$ and the eutectoid structure $Fe_3O_4$/Fe. The obvious difference from No. 1 group is the existence of the eutectoid structure $Fe_3O_4$/Fe due to the short storage time in the air.

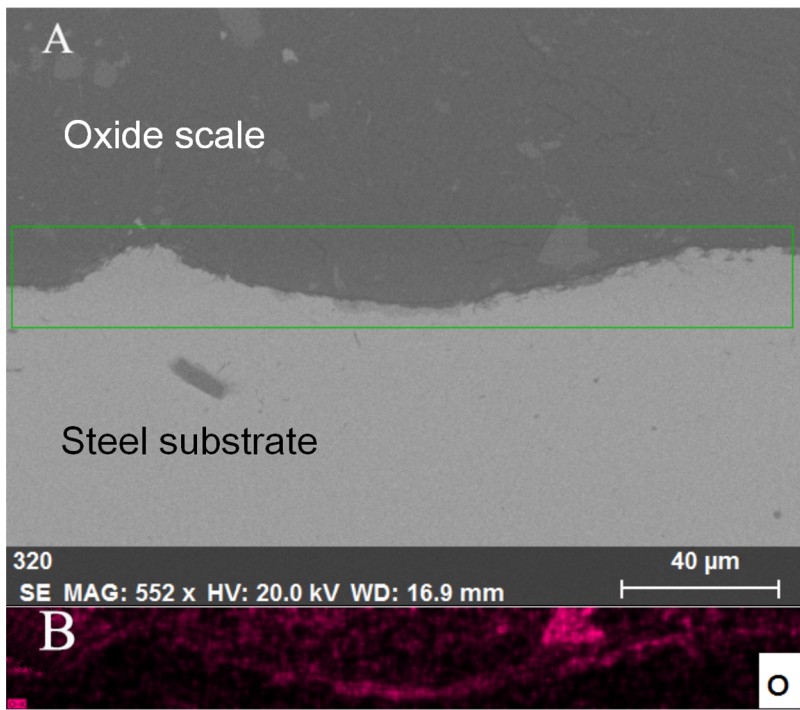

**Figure 12 Results of oxygen element scanning of No. 2 group (WD = 16.9 mm).** (A) The part in the green frame is the area scan object; (B) oxygen scan results. A larger view field chosen for the area scanning of energy dispersive spectrometer for No. 2 group is shown.

## Analysis of the shot blasting experiments with a small amount of abrasive

### Analysis of the descaling effects from the front surface's scanning

As is shown in Fig. 5A, for the No. 1 group, at the edge of the hitting pit, a large area of the oxide scale fall off, and a large number of cracks appeared on the surface of the remaining scale layer. The peeled areas are large and the descaling effect is good. As is shown in Fig. 5B, for the No. 2 group, only a few oxide scale fall off at the junction of the hitting pit edge, and there are only a few tiny cracks on the remaining oxide scale layer. The peeled areas are small and the descaling effect is worse compared with the No.1 group.

### Analysis of the descaling effects from the section's scanning

As shown in Fig. 6A, for the No. 1 group, after the shot blasting with a small amount of abrasive, the oxide scale at the pit's edge fall off completely and the basal body is revealed and the peeled areas are large. There are not obvious cracks of the oxide scales in and around the pits, but there are tiny gap between the scale layer and the basal body near the peeled areas.

As is shown in Fig. 6B, for the No. 2 group, the peeled areas of the scale layer is small, and the basal body is not completely revealed. However, there are obvious cracks of the oxide scales in the pits. Thus, it can be deduced that compared with No. 1 group, the oxide scale of the specimens of No. 2 group has lower hardness and better combination

with the basal body. The descaling effect of No. 1 group is better when the impact force of the projectile reaches a certain level.

## Analysis of the shot blasting experiments with a large amount of abrasive

### Analysis of the descaling effects from the front surface's scanning

Figure 7A shows the descaling effect at 100× magnification in the backscattering mode. The darker part represents the area where the oxide scale has not fallen off, and the lighter part represents the area where the oxide scale has fallen off. It can be seen that most of the oxide scale has been peeled and only a few remains after the shot blasting with a large amount of abrasive. The 500 times magnification of the peeled areas is shown by Fig. 7B, and it can be observed that the pits of the basal body have become relatively smooth due to multiple hits.

Figure 8A shows the descaling effect at a magnification of 50 times, and Fig. 8B is a partial enlarged view of Fig. 8A. The relatively uniform color in the figures indicates there is only one kind of material in the surface. And the EDS results show that the oxygen content is 30.28%, the iron content is 69.72%, which indicates that the layer is the remaining oxide scale rather than the basal body. It can be obtained that the outer oxide scale layer falls off during the shot blasting process, but the inner oxide scale layer still exists on the substrate, which also confirms that the oxide scale is a layered structure.

### Analysis of the descaling effects from the section's scanning

As is shown in Fig. 9, the oxide scale after the shot blasting with 20 kg of abrasive for No. 1 group has been peeled cleanly without obvious residue, and the surface is smooth after a large number of random hits. As is shown in Fig. 10, the oxide scale after the shot blasting with 20 kg of abrasive for No. 2 group has not been peeled completely, but the thickness is reduced from 12 μm to 6 μm, which means that the outer oxide scale falls off with the shot blasting, but the internal scale layer still exits. In addition, obvious cracks appeared on the surface of the remaining oxide scale.

As is shown in Fig. 11, where a larger view field was chosen compared with Fig. 9, the result of the area scan can indicate the content of the element by the depth of the color. The scanning area is shown by the green line frame in Fig. 11A, and the scanning result of oxygen element is shown in Fig. 11B. It can be seen that there is no large amount of oxygen between the mounting powder and the basal body, which indicate that the oxide scale has fallen off after a large number of shot blasting and there is no oxide scale remaining.

As is shown in Fig. 12, where a larger view field was chosen compared with Fig. 10, the scanning area is shown by the green line frame in Fig. 12A, and the scanning result of oxygen element is shown in Fig. 12B. It can be seen that there is a significant area of oxygen accumulation between the mounting powder and the substrate, which indicates that after a large amount of shot blasting, the oxide scale still exists.

It can be known from the above experiments that the difficulty of oxide scale removal is related to the content of $Fe_3O_4$ in it. For steel strip that has been stored for a long time,

the main components of the oxide scale are $Fe_2O_3$ and $Fe_3O_4$, and the oxide scale can be more easily removed by shot blasting; while for the steel strip with shorter storage time, the oxide scale contains $Fe_3O_4$, shot blasting can reduce the thickness of the scale layer, but only much longer shot blasting time can make the oxide scale completely fall off.

For oxide scale without eutectoid structure, in the case of only descaling by shot blasting, as the thickness of oxide scale gradually decrease, the efficiency of descaling will be greatly reduced, resulting in increased costs. Therefore, after the shot blasting and descaling, an additional high-pressure water jet process can be added. Firstly, a large area of oxide scales is removed by shot blasting. At this time, the binding capacity between the remaining oxide scales and the basal body becomes weak, and then it can be completely removed by direct spraying with high pressure water further.

For oxide scale with eutectoid structure, using shot blasting to remove oxide scale is less effective. The method of combining shot blasting and pickling should be explored. By studying the best process, it can reduce pollution emissions and production costs and improve production efficiency.

## CONCLUSIONS

In this article, two kinds of Q235 strips stored at different times were selected to analyze the difference of surface oxide scale composition and the effect of shot blasting descaling, which provided a basis for the optimization of shot blasting process. The main research contents and conclusions are as follows:

1. The EDS and XRD were used to observe and analyze the composition of the two Q235 steel scales stored at different times. It is found that the composition of the steel strip after hot-rolling is significantly different during long-term storage. During the storage of the strip, the oxide scale will continue to be oxidized, and the eutectoid structure $Fe_3O_4/Fe$ of the inner layer will be oxidized to $Fe_3O_4$. The hot-rolled strip scale with long storage time will have no eutectoid structure $Fe_3O_4/Fe$ and FeO.

2. The descaling experimental facility designed by NERCFRE was used to perform shot blasting and descaling treatment. The scanning electron microscope was used to observe the effect of a small number of shot blasting effects of two Q235 strip steels. Although $Fe_2O_3$ and $Fe_3O_4$ have high hardness, they are easy to fall off during shot blasting, and the strips that have not been stored for a long time are prone to scaly fracture due to the presence of $Fe_3O_4/Fe$ eutectoids. However, it is more firmly bonded to the basal body, and it is relatively difficult to remove the oxide scale.

3. The scanning electron microscope was used to observe the effect of a large number of shot blasting effects of two Q235 strip steels. It is found that for strips that have been stored for a long time, the main components of the oxide scale are $Fe_2O_3$ and $Fe_3O_4$, which can be more easily removed by shot blasting; while for strips that have been stored for a short time, the scales contain eutectoids structure $Fe_3O_4/Fe$, shot blasting can reduce the thickness of the oxide scale, but it is more difficult to completely remove it.

4. According to the experimental analysis in this article, it is found that due to the presence of the eutectoid structure $Fe_3O_4/Fe$ in the oxide scale, it is more difficult to remove

the oxide scale. The subsequent research should adjust the shot blasting descaling process for different oxide scale components, such as the combination of shot blasting and high-pressure water direct injection. At the same time, it is also possible to explore the descaling process combined with pickling and find the optimal ratio of shot blasting descaling and pickling to achieve the comprehensive optimization of reducing pollution emissions, reducing production costs and improving production efficiency.

### Funding
This work was supported by the National Natural Science Foundation of China (Grant Nos. 51975043 and 51604024) and the Supported by Beijing Natural Science Foundation (3182026). The funders had no role in study design, data collection and analysis, decision to publish, or preparation of the manuscript.

### Grant Disclosures
The following grant information was disclosed by the authors:
National Natural Science Foundation of China: 51975043 and 51604024.
Beijing Natural Science Foundation: 3182026.

### Competing Interests
The authors declare that they have no competing interests.

### Author Contributions
- Xiaochen Wang conceived and designed the experiments, analyzed the data, performed the computation work, prepared figures and/or tables, and approved the final draft.
- Rui Ai conceived and designed the experiments, analyzed the data, performed the computation work, prepared figures and/or tables, and approved the final draft.
- Quan Yang conceived and designed the experiments, authored or reviewed drafts of the paper, and approved the final draft.
- Shang Wang performed the experiments, analyzed the data, prepared figures and/or tables, and approved the final draft.
- Yanjie Zhang performed the experiments, performed the computation work, authored or reviewed drafts of the paper, and approved the final draft.
- Yingying Meng performed the experiments, performed the computation work, authored or reviewed drafts of the paper, and approved the final draft.
- Xianghong Ma analyzed the data, performed the computation work, authored or reviewed drafts of the paper, and approved the final draft.

### Data Availability
The raw data are available in Tables 1–2 and Figures 1–12.

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
