# Peer review of "Effect of oxide scale structure on shot-blasting of hot-rolled strip steel"

_PeerJ Materials Science, doi:10.7717/peerj-matsci.9_

## Round 0.1 · original submission · Major Revisions

· Academic Editor

Major Revisions

The paper ended being reviewed by three people and they have a variety of comments. They need to be addressed; I believe the paper is publishable after the comments have been addressed.

Reviewer 1 ·

Basic reporting

In this manuscript, the authors investigated the effect of oxide scale composition of hot-rolled strip steel on shot blasting.

Experimental design

An innovative acidless descaling technology has been proposed using slot blasting and high-pressure jet.

Validity of the findings

The research findings are interesting and significant, and the discussion has been well organised.

Additional comments

In this manuscript, the authors investigated the effect of oxide scale composition of hot-rolled strip steel on shot blasting. An innovative acidless descaling technology has been proposed using slot blasting and high-pressure jet. The research findings are interesting and significant, and the discussion has been well organised. However, this manuscript requires revision based on the comments below.
1. Please indicate resin, oxide scale and steel substrate in all the SEM images.
2. In Fig. 3, the thickness of oxide scale in sample groups 1 and 2 is quite different. After using the abrasive under the same weight and shot blasting time, it is easy to understand that sample 2 has more oxides retained due to thicker oxide scale. Except for oxide scale composition, other influences such as hardness and compactness of oxide scale on shot blasting should also be considered.
3. In Fig. 4, please correct the Chinese characters in Y axes.
4. In Fig. 5, it is hard to distinguish the different amounts of cracks after shot blasting with 2 kg of abrasive in sample groups 1 and 2. Please indicate the cracks in the SEM images.
5. In Fig. 6, please use the same magnification in the images for a clear comparison. To see the cracks more clearly, it is better to add enlarged images focusing on the interface of oxide scale/steel substrate.
6. In Figs. 7 and 8, please use consistent magnifications to well compare the descaling effects in sample groups 1 and 2. Please also indicate the exact zones in Figs. 7A and 8A for an enlarged observation in Figs. 7B and 8B, respectively.
7. In Figs. 10 and 12, it is better to use consistent magnification to compare the EDS mapping of Oxygen.

Reviewer 2 ·

Basic reporting

Overall, this is an interesting inquiry into steel oxidation, how storage time affects the structural evolution of the oxide scale, and how the structure impacts the optimal descaling approach. It needs significantly more detail in the methods to support its conclusions, and perhaps some additional thought on whether observations are attributable to composition or morphology.

Language use is decent. Needs some work in a few areas, see general comments section for specifics. Title should be reorganized, if possible, for readability and clarity – I recommend “Effect of oxide scale structure on shot-blasting of hot-rolled strip steel.”

The introduction needs a little bit of readability work; but the literature cited seems relevant and the establishment of the background – oxide scaling approaches and their uses and limitations – is strong and provides good detail. In several places the authors say that “research is not sufficient” on particular topics. It would strengthen the paper to explain how this insufficiency impacts productivity, and/or how this insufficiency will be addressed by the present research.

Most figures are of acceptable quality. Interpretation and results could be improved through the inclusion of some additional data, specific recommendations below. Scale bars are included on all images. I believe there is a repeated typographical error in Figure 1, I recommend the authors carefully review figures and tables to catch any other such errors. I would like to see significantly more detail on SEM and XRD settings either in the captions or in the main body of the text, as well as details on sample preparation. The omission of these elements considerably weakens the conclusions – see section 3 of this review.

Experimental design

The introduction could do a better job of framing the research question. It was clear by the end of the paper what the question was, but I would have liked a clearer picture at the outset of why the “storage time” or “air-cooling time” is relevant in the broader scope of steel oxide layer removal. Would it be better for the authors’ stated goals of efficiency and lowering emissions to have a short storage time or a long one, perhaps considering the manufacturing and supply chain? More discussion on this topic would help frame the purpose of the paper for the reader.

My concerns about rigor and detail are constrained to the analytical methods and sample preparation, which I will cover in detail in the following sections.

Replication of these findings will be difficult without additional information on the samples themselves as well as the analysis methods. A thermal history of the steel samples during hot-rolling would help. This is partially addressed in the introduction, but never fully explained. A simple figure showing representative dwell temperature vs. time for Q235 during the hot-rolling process would make the article more accessible to the general materials science audience, not just those versed in steel manufacturing.

Validity of the findings

My main concerns with this paper have to do with the robustness and controlled nature of the data presented. I believe the data is legitimate and supports the authors’ conclusions (with some exceptions, see Section 4), but because these conclusions depend purely on interpretation of SEM images and XRD phase IDs, much more background is needed. General improvements for presented figures:

• XRD data should include:
o Sample preparation & mounting
 Were samples powdered/pulverized? How? How much was analyzed?
• If not, how were the samples mounted? How were orientation effects filtered from the data?
o Instrument model & make
o Tube/anode material and/or x-ray wavelength
 Tube excitation parameters (KV/mA) would also be preferable
o Calculated/literature phase data (“stick patterns”) and source thereof (ICDD, COD, card number, etc.)

• SEM data should include:
o Sample preparation (sectioning, mounting)
o Accelerating voltage
o Working distance
o Imaging mode (SEI, BSEI)
 If a backscattered electron image (BSEI):
• Imaging mode (i.e. compositional vs. topographical)
• Tilt angle

This information can be supplied either in the Experimental Methods section or in the figure captions. Detailed review on individual figures can be found in the general comments.

I believe the compositional assertions about the eutectoid phase must be identified as speculation, without the inclusion of additional data or literature support.

Additional comments

Figures

Figure 1. Desaling -> Descaling

Figure 3. Labeling the figures with their approximate air exposure time (1 year vs. 1 week) will help readability. Combining this figure with some of the later cross-sectional analysis may also help readability.

Figure 4. In addition to the general notes from section 3: the y-axis should be in English, I am assuming this is “Intensity (counts)” or similar. The data could use a background correction, as there is some scattering contributing to the background in the 10-30 2θ region. Additionally, there are some phase-identification discrepancies that would be resolved by the inclusion of the literature patterns – with ICDD or COD card catalog numbers – on the figure itself. This typically requires reproduction in color, so alternative presentations may be necessary if color is not available.
Peak ID inconsistencies that need correction:
 30° 2θ – M, O, or both?
 38° 2θ – M or O?
 58° 2θ – M, O, or both?
 90° 2θ – M or I?
The primary issue with this figure is the claimed ID of a F3O4/Fe phase. The data as presented show that the authors have observed an Fe phase (having a tight correlation with Q235 literature patterns) and a magnetite (Fe3O4) phase in both samples. However, in order to identify a eutectoid (i.e. a solid solution of two phases) more careful analysis of peak shifts and peak intensities would have to be performed, as the interatomic distances should change as a result of varying Fe inclusion, compared to the parent phases. For the record, I do not believe that the presence or non-presence of the eutectoid phase is necessary to explain the differences in behavior of the oxide scale, so references to this phase could be removed, the inconsistencies above could be corrected, and the data could be presented as-is.
The important difference I do see is that Figure 4B shows a higher percentage of the Fe phase, but without further explanation of the samples’ preparation and mounting, there is no way to tell that the variance in the Fe phase is not simply due to variation arising from the presumed pulverization/grinding process, or possibly due to the thicker deposit of oxides (i.e. higher mass fraction) in the No. 2 group. Inclusion and discussion of the sample preparation would strengthen the central claims about the different oxidation times requiring different descaling methods.

Figure 5. This figure could be improved by adding a reference surface, i.e. before blasting.

Figure 6. This figure could be combined with Figure 5, so that the surface and cross-sectional views are next to each other. The claims about No.2/6B having “better combination with the basal body” in line 199 could use additional support, this does not seem obvious to me from the image. An alternate theory could simply be that the No.1 film is more cohesive and densified, therefore when shot-blasted, it comes off in larger flakes rather than being crushed into smaller pieces. A higher magnification image or, better, a compositional BSE image showing a density gradient in this zone might provide evidence of this “better combination” claim. HRTEM would provide conclusive proof, but these experiments are difficult and lengthy.

Figure 7. Additional notes about the imaging modes are needed – 7A is clearly a compositional BSE image but 7B may be topographical BSE, or even SEI. SEM firing parameters and working distance are needed to allow for replication.

Figure 8. The claim about “color uniformity” on lines 213-214 is only true if this is a compositional BSE image. Please specify, the contrast variations at edges indicate that these are SE images, which carry no information about composition. Additionally, if these are compositional BSE images, these contrast variations will need to be addressed in the text.

Figures 9-12. I would recommend combining these for direct comparison – figures 9 and 11 can be combined so that they can be compared side-by-side. Same with 10 and 12. The imaging parameters included in figures 10/12 should be included for other images in the manuscript, though usually these are in the caption or the Experimental Method section. The data would also be stronger if EDS mapping of Fe for these regions were included. Also – figure 9/11 does a better job (but still not entirely conclusive) demonstrating the “better combination” claim from Figure 6. This is the type of magnification needed, preferably with before-and-after images.

Table 1. How was the chemical composition determined?

Table 2. How was position in the sample determined? Was this paired with a depth-profiling technique? Details are needed!

Text

Line 19 – explain what “the related researches are not sufficient” means
Line 33 – specify that we are talking about steel.
Line 45 – “impact of water is not sufficient” – specify, for what?
Line 72-73 – English usage
Line 90 – “research… on the blasting descaling effect is not sufficient” – specify what information is missing – what do we not know that this paper will address?
Line 106-107 – include thermal history of the samples here
Lines 116-147 – include sample preparation, analysis method settings, additional detail from figure commentary here
Line 131 – unless an equation is going to be shown using these parameters, symbols are not necessary
Lines 154-155 – Why does the sample with the longer oxidation time have a thinner scale? I would like to see some discussion of density here, not just composition.
Line 157 – this “well combined” claim shows up several times, it is not obvious from the data presented. Either provide more analytical support or remove.
Line 216-218 – the behavior described here could also result from poorly-densified, morphologically heterogeneous oxide crystals, rather than a layered structure. Layering should be clear from the cross-sectional SEM in Figure 3, but it simply looks disorganized and poorly densified. I think this is the real conclusion from the data – somewhat counterintuitively, well-formed, highly dense oxide scales are easier to shatter and remove under shot-blasting.
Line 225 – don’t state that results are accidental. Just describe what is learned from the high-magnification image, and what is learned from lower-mag/EDS.
Lines 246-252 – I think this will need to be rewritten, unless the data more conclusively demonstrates the presence of this eutectoid phase.
Lines 253-259 – it seems odd to spend much of the paper showing how the No.1 oxide scales are more completely removed and then to turn around here and say that the descaling will be less efficient and more expensive. I am not sure this is what the authors intend.

·

Basic reporting

I felt that the written structure of the article generally followed acceptable practice

Experimental design

The abstract suggested that the properties of the scale would explain the effectiveness of the descaling process.

I was disappointed by the very limited experimental design of the investigation which only looked at two aging conditions of the steel and its surface scale which was 1) no aging and 2) aging at ambient after one year. I cannot believe that manufacturers or consumers would want to wait for a year to be able to effectively descale these materials so that they could avoid various other methods including pickling.

This work did not directly compare shot blasting and pickling to show the difference. In the methods section, the experimental equipment was described where the process parameters were detailed to include 1) angle of impingement, 2) velocity, 3) shot diameter and 4) amount of shot used. Only the last variable was apparently used, and no other attempt was done to determine it other factors might have contributed to the success of the investigation. There was no optimization of the process claimed in the conclusions because there were no other factors that were tested. There was only one type of shot blasting material used.

Validity of the findings

I was looking forward to see the properties of the scale explained by the morphology and chemical composition, however it did not go far enough. Only the average chemical composition of the scale was used as a characterization. The Fe / O phase diagram was referred to in the quoted references, but was not discussed in enough detail to establish what was expected in the present investigation. Obviously the scale present in fresh material was not in equilibrium, and it was implied that material stored for a year was nearly in equilibrium. If it was important to the success of the investigation to produce the equilibrium phase and distribution, why then was no attempt made to accelerate the formation of the equilibrium product?

While the morphology was shown in layer thickness, there was no quantitative expression of grain size or other distribution of the phases present. No attempt was made to assess the mechanism of adhesion or bonding or the strength of the adhesion of the scale layers to help explain the observed behavior. I would imagine that a micro shear test or bend tests as described for tension descaling would be appropriate methods to derive the properties of the scale needed for this characterization.

The introduction reviewed an extensive list of competing descaling processes used in the industry compared to shot blasting, but no attempt was made to make an assessment of the relative cost, material aging or logistical tradeoffs between these methods. There was also no pass / fail criteria expressed to help judge the ultimate success of failure of the results.

---

## Round 0.2 · Minor Revisions

· Academic Editor

Minor Revisions

Please address the remaining reviewers' comments.

Reviewer 2 ·

Basic reporting

Rewritten sections are improved and clearer. I am satisfied with the changes, with the exception of the XRD data, see section 2.

Experimental design

I do not feel that my commentary on the XRD data has been appropriately addressed.

The authors indicate the samples are simply attached to the XRD sample holder and scanned, without any processing. This results in the potential for changing intensities due to orientation of the crystallites within the sample. Generally samples are powderized before XRD analysis. Without this step, there is no way to be assured that the entire pattern of any particular phase is appearing, because the method for phase identification depends on having the analyzed crystallites randomly oriented within the sample volume. In the specific case here - an oxide film growing off of a flat substrate - it is likely that grains will grow in a particular direction, resulting in preferred (and unknown) orientations within the sample. It introduces many questions about the data. This should be addressed in the experimental method - how these data were refined to address the orientation questions. If they were not refined, they should be.

In the rebuttal letter, the authors mention PDF cards. I'm assuming this means ICDD data - please include the card numbers for the specific patterns used somewhere in the article.

The authors say the X-ray tube is "ceramic" - this may be the housing, but the actual anode material is, in virtually all cases, a metal target. What I am really asking for here is which wavelength(s) the samples were illuminated with. If it is standard Cu k-alpha (1.541 angstroms) this should be stated somewhere. The name of the instrument is insufficient, most suppliers offer many tube options, though 95% of the instruments have Cu anodes. This is critical, along with the PDF numbers, for repeatability and comprehension of the data.

Finally, the peak identifications themselves are inconsistent from Figure 4A to Figure 4B. This may be a result of the orientation issues mentioned above, which can cause peaks in literature powder patterns to not appear in scans. However, if that is the case, additional data must be presented to justify assigning a peak to one pattern in one sample and a separate pattern for the second. The most egregious cases are, again, as noted in my previous review:

30° 2θ - assigned to both M/O in 4A, assigned to O in 4B.
38° 2θ - assigned M in 4A, assigned O in 4B.
58° 2θ - assigned to both M/O in 4A, assigned to M in 4B.
90° 2θ - assigned I in 4A, assigned M in 4B.

The authors claimed to address these in the rebuttal, but the uploaded images have not been changed. Additionally, they simply assign 3 of the peaks to both patterns (highly improbable, based on what I've seen of Fe3O4 and Fe2O3 patterns in the literature), and I believe they misidentify the peak at 90 as Fe - I suspect the actual Fe peak is at 82, again, based on my own very limited searching of the literature. Lastly, the literature patterns themselves are not shown in the image nor referenced in the document. There is additionally a prominent peak at 33° 2θ assigned to phase O in 4A that disappears in 4B, despite still identifying the O phase for that sample. If the authors simply include the literature patterns (as "sticks" underneath the scan itself, or in a separate chart) on the image, these questions are resolved, and the phase ID will be more-or-less obvious.

The reason the discussion above (on powders vs. bulk) is relevant here, is that it is not easy to determine that the disappearance of the peak at 33 is because the sample lacks Fe2O3, or because the Fe2O3 crystals are oriented in such a way that this peak is not revealed in that sample. Similarly the changes in intensity for the I phase may be related to preferred orientation, as the ratio between the two peaks does appear to change from sample to sample - it is simply not possible to know, without further data treatment and explanation in the paper.

Finally, the eutectoid phase mentioned still has no literature pattern nor phase ID. I am not familiar with this subfield, so "eutectoid Fe/Fe3O4" may be a misnomer, but generally, this refers to a solid solution, not the mere simultaneous presence of two distinct phases (this would be something like "heterogeneous polycrystal" not eutectic or eutectoid). In my opinion, it may be better to rely more (if not entirely) on the EDS data from Table 2 to make the point about the oxygen composition of the scale and how it affects shot peening.

On the other hand, I feel the electron microscopy work has been clarified well, and have no further issues.

Validity of the findings

See above critique of the XRD data. Otherwise, I feel the paper is valid.

---

## Round 0.3 · accepted · Accept

· Academic Editor

Accept

Thank you for addressing the comments. The paper has been accepted.